# Cross-Sectional Correlational Study in the Valencian Community (Spain) on the Social Image and Attitudes Towards Nursing

**DOI:** 10.3390/healthcare13222834

**Published:** 2025-11-08

**Authors:** Silvia Solera-Gómez, David Sancho-Cantus, Jesús Privado, Jorge Casaña Mohedo, Cristina Cunha-Pérez

**Affiliations:** 1Francisco de Borja Hospital, 46702 Gandía, Spain; solera_sil@gva.es; 2Doctoral School, University of Valencia, 46010 Valencia, Spain; 3Department of Nursing, Faculty of Medicine and Health Sciences, Catholic University of Valencia, 46007 Valencia, Spain; david.sancho@ucv.es (D.S.-C.); cristina.cunha@ucv.es (C.C.-P.); 4Department of Methodology of Behavioral Sciences, Universidad Complutense de Madrid, 28040 Madrid, Spain; jesus.privado@pdi.ucm.es; 5SONEV Research Group, School of Medicine and Health Sciences, Catholic University of Valencia, 46007 Valencia, Spain

**Keywords:** social image, nursing, attitude towards nursing, empathy, professional values, communication skills, cross-sectional study

## Abstract

**Background:** Nursing is an essential pillar in health services provision; however, its social value is often underestimated. The public image of, and society’s attitude toward, the profession is commonly influenced by stereotypes and biases. Objective: This study aimed to analyze the predictive influence of empathy, professional values and communication skills on the social image and attitude towards nursing. **Methods:** A cross-sectional, correlational study was conducted in the Valencian Community, Spain. Snowball sampling was used for data collection from 300 participants (81% female; mean age 35.85 years, SD = 14.99). Empathy, professional values and communication skills were measured, and a structural equation model was proposed to assess their influence. **Results:** Professional values were significant predictors of both social image (β = 0.41) and attitude toward nursing (β = 0.34). Similarly, communication skills predicted social image (β = 0.31) and attitude (β = 0.37). Empathy also emerged as a significant, though minor, predictor. Collectively, these three factors explained 30% of the variance in social image and 39% in attitude toward the profession. The main limitations arise from the severe demographic bias of the snowball sample (skewed toward women, young, and highly educated individuals) and the modest explanatory power (R^2^ of 30–39%). This limits the generalizability of the findings and suggests the need for future research on omitted variables, such as working conditions and organizational culture. **Conclusions:** Empathy, professional values and communication skills are key competencies contributing to a more positive social image of and attitude toward nursing. Investing in the development of these competencies can significantly enhance the recognition and appreciation of nursing within the healthcare system.

## 1. Introduction

Nursing plays a vital role in contemporary society; however, the value it brings to patient and family care is still often unrecognized [1]. Moreover, the profession contends with an ambiguous, devalued image perpetuated by stereotypes and biases [2]. The general public often fails to appreciate the leadership capacity nurses exercise across their functions, including management, quality care provision, research to update procedures, and teaching. Understanding society’s perception of nursing is essential, as this view demonstrably influences professional performance and quality of care [3]. Strategies to improve the social image must therefore begin by exploring how nursing is conceived and what factors determine this perception [4].

The public perception of nursing remains a top priority, as society often holds a distorted or ambiguous image of the profession [5]. This social image is inextricably linked to factors like professional reputation, job satisfaction, the quality of care provided, and the overall well-being of individuals and communities [6,7]. The negative aspects are frequently perpetuated by stereotypes, most notably gender bias [8], which historically associates nursing with women and maternal care [3]. This external evaluation affects professional self-concept, which is vital for development and well-being [9] and is influenced by general perception of the profession and effective communication styles [6].

The most prominent predictors of the social image of nursing are empathy [10], professional values [11], and communication skills [12]. Empathy, emotional intelligence, and problem-solving are considered essential skills for 21st-century nursing [13]. In fact, empathy is considered a core psychological competence for caregiving [14], encompassing perspective-taking and compassionate care [15]. High empathy scores predispose **individuals** to higher levels of personal satisfaction [16], which influences their professional self-perception [17]. Conversely, low empathy scores correlate with decreased professional competence, which negatively affects self-image [15].

Professional values, based on the inherent characteristics of a profession, serve as an element to shape the identity of a professional group like nursing. Following the conceptual analysis by [18], nursing professional values have been defined by [19] as Human Dignity, Integrity, Altruism, and Justice. These are crucial professional principles and frameworks for clinical practice and outcome assessment. Thus, nursing professional values are closely related to factors concerning individuals, the profession itself, and society. The professional identity of nursing is, in turn, closely linked to factors such as professional reputation, workplace, and professional values [2].

The influence of certain nursing stereotypes on student perception and value formation during career choice has been widely studied. Recent studies [20] suggest that current nursing education should specify the stage at which different learning styles are developed and professional values are consolidated. Therefore, the cultivation of strong professional values can enhance not only ethical decision-making but also the social image of nursing [21,22].

Effective communication between individuals and healthcare professionals is a critical tool for accurate decision-making, improving outcomes, and meeting the needs of individuals and families. To achieve effective communication, several components are essential, such as active listening, empathy (as previously mentioned), and ethical awareness, which align with professional values [23].

Recent studies confirm that care behavior is directly related to communication skills [24]. Furthermore, the Joint Commission International [25] attributes 80% of medical errors to poor communication. Consequently, a deficit in communication skills among healthcare professionals directly compromises the quality and safety of care provided to individuals and the healthcare system as a whole. Communication skills are one of the six general competencies defined and adopted by the Accreditation Council for Graduate Medical Education and the American Board of Medical Specialties, thus providing guidance on their importance in the clinical healthcare setting [26].

Nurses who have a better self-image also demonstrate better communication skills, which increases personal satisfaction and enhances professional values, thereby reinforcing professional identity [27]. Collectively, social skills (including communication and empathy) and professionalism (which encompasses professional values) are hypothesized to predict the image of nursing in society [28,29].

The literature suggests that the same three elements significantly predict the attitude toward nursing: empathy, professional values, and communication skills [30]. For instance, compassion in care directly impacts the public’s attitude toward nursing, influencing professional values. More empathetic professionals tend to be treated with greater respect and are perceived as more professional. These professional values, encompassing ethics and commitment, subsequently influence the attitude toward nursing by shaping certain stereotypes and the perceived professional roles. Furthermore, communication skills (including empathy) play a key role in the attitude shown by society and are related to other elements like professional values.

Based on this rationale, the hypothesis of this study is as follows: Empathy, professional values, and communication skills are positively correlated and significantly predict both the social image of nursing and a favorable attitude toward the profession. The way the population perceives nursing is extremely relevant, as it plays a fundamental role in the recognition and visibility of the profession [3]. Furthermore, understanding the factors that influence a more positive attitude toward the profession is critical. Therefore, the aim of this study was to examine the extent to which empathy, professional values, and communication skills influence the social image of and attitude toward nursing.

## 2. Materials and Methods

### 2.1. Participants

The final sample consisted of 300 participants. The mean age was 35.85 years (SD = 14.99), and 81.0% were women. The educational distribution was as follows: 1.3% reported primary education, 1.0% secondary education, 19.0% high school, 23.0% vocational training, and 57.7% university education.

### 2.2. Instruments

The Nursing Image Scale (NIS) [31] measures the image of the nursing profession. It comprises 35 items on a 3-point Likert scale and assesses dimensions such as general impression, communication, working conditions, level of training, and suggestions related to the profession. The scale was previously validated for use in Spain [30]. The internal consistency (Cronbach’s alpha) obtained in the current sample was 0.77, exceeding the recommended minimum threshold of 0.70 [32].

The Nursing Attitude Questionnaire (NAQ), designed by Toth et al. [33], assesses attitude toward nursing. The initial version comprised 30 items measuring nursing functions, values, professional activities, and responsibilities. The items are scored on a 5-point Likert scale, where higher scores indicate a more favorable perception. The questionnaire was validated in Spanish [30]. The internal consistency (alpha) for the current sample was 0.79.

The Jefferson Scale of Physician Empathy-Healthcare Professionals (JSE-HP) [34] was used to measure empathy. It consists of 20 items answered on a 7-point Likert scale and captures three dimensions: taking the patient’s perspective, compassionate care, and putting oneself in the individual’s place. The version tailored for health professionals was utilized, demonstrating an internal consistency (alpha) of 0.81 in this study.

The Nursing Professional Values Scale (NPVS) [35] includes 26 items scored on a 5-point Likert scale. It is structured according to the American Nurses Association Code of Ethics and measures five factors: caring, activism, trust, professionalism, and fairness. The internal consistency (alpha) for the present test was 0.90.

The Scale of Communication Skills in Health Professionals (EHC-PS) [36] is composed of 18 items rated on a 6-alternative Likert-type scale. It presents four dimensions: Informative Communication, Empathy, Respect and Social Ability. The internal consistency (alpha) obtained a value of 0.89.

### 2.3. Procedure

The eligibility criteria for participants were established to ensure response quality and comprehensibility. The inclusion criteria required that participants be over 18 years of age and possess an adequate command of the Spanish language. Furthermore, they should not have cognitive limitations that could hinder their ability to understand and accurately complete the questionnaire.

The snowball sampling method was employed, wherein the different instruments were administered via Google Forms disseminated through social networks. Respondents were asked to share the survey with other individuals. Through this tool, participants were provided with information regarding the study characteristics, informed of their right to withdraw from the study at any time without consequence, and guaranteed anonymity and confidentiality. Participants took an average of 15 min to complete the questionnaires.

### 2.4. Ethical Considerations

The study was conducted in accordance with the principles of the Declaration of Helsinki [37]. Approval was obtained from the Human Research Committee of the University of Valencia (procedure nº 2023-ENFPOD-2638296). All participants included in the study provided informed consent after being duly informed about the procedures and nature of the research.

### 2.5. Data Analysis

Data analysis proceeded in three primary steps. First, the distribution of the applied measures and their internal consistency were analyzed. Second, Pearson correlations were obtained for all study measures. Third, a structural equation model (SEM) was estimated to determine the predictive weight of empathy, professional values, and communication skills on social image and attitude toward nursing.

The sample size was deemed adequate based on the guideline recommending ten participants per indicator for factor analyses [38]. With 300 participants and 20 indicators in the most complex model, the resulting ratio of 15:1 is considered adequate. Furthermore, a power analysis (1 − β) for the employed sample size was calculated assuming a significance level of 5% for a medium effect size correlation (r = 0.30) according to Cohen [39]. The power obtained was 0.998, which clearly exceeds the recommended criterion of 0.80.

To assess the fit of the data to the contrasted model, three types of goodness-of-fit indices were employed. Absolute Indices, which test whether the theoretical model fits the empirical data, included the ratio of the chi-square to degrees of freedom (χ^2^/df) [40] (values below 3 indicate a good fit); the Goodness-of-Fit Index (GFI) [41] (values > 0.95 considered a good fit); and the Standardized Root Mean Square Residual (SRMR) [42] (values < 0.08 indicating a good fit) [32]. Incremental Indices, which compare the obtained model with the null model, included the Normed Fit Index (NFI) [38] (values > 0.95 indicate a good fit). Parsimonious Indices, which penalize the number of estimated parameters, included the Parsimony Goodness-of-Fit Index (PGFI) [39] and the Parsimony Normed Fit Index (PNFI) [43] (both with values > 0.50 indicating a good fit).

Given the absence of multivariate normality, the Unweighted Least Squares () estimation procedure was employed, as recommended for such conditions [32,38]. Moreover, the possibility of multicollinearity was assessed by examining correlations between the predictors. Since the highest correlation was 0.67 (well below the critical threshold of 0.80 for concern) [32], multicollinearity was not deemed a threat to the regression coefficients. Analyses were performed using the SPSS V. 23 statistical package and the AMOS V. 23 program [44].

## 3. Results

A severe demographic bias was detected in the sample, resulting in a lack of social representativeness. Snowball sampling yielded an unbalanced sample, composed predominantly of women (81.0%), relatively young individuals, and participants with a high level of education (57.7% university graduates) [30]. This skewed distribution makes it impossible to ensure that perceptions of the image and attitude toward nursing reflect those of society in general. Furthermore, the lack of stratification analysis by age, gender, and educational level failed to mitigate this risk, suggesting that the results are specific to a particular demographic subgroup rather than generalizable.

### 3.1. Descriptive and Correlations

Table 1 shows the descriptive statistics of the different measures used in this study. As shown, all measures, except the Professionalism subscale (NAQ), exhibited adequate values of skewness (≤|±2.00|) and kurtosis (≤|±7.00|) [45].

Furthermore, the Pearson correlation between the different measures was calculated (see Table 1). Taking a minimum correlation of |±0.30|as the reference for a mean effect size, according to Cohen [39], the results indicate numerous correlations with a medium-to-high effect size. This suggests that the measures share common variance, which supports the creation of a predictive model based on common factors.

In terms of assessing the sample’s representativeness, it is worth noting that currently, according to the Health Report presented in 2024, 85.5% of nursing professionals are women [46]. The values obtained regarding educational level reflect a similarity with the data from the Spanish National Institute of Statistics (INE), which shows that 22% have a secondary education level and 42.3% have a higher education level [47]. This yields values very similar to the population sample used in this study. Furthermore, the age range considered in the study could be considered representative, given that the annual social media study by IAB Spain reveals the average age of users is 43, an age consistent with that obtained in our study, considering the standard deviation of around 15 years [48].

### 3.2. Predictive Model

A predictive model with five latent factors was estimated to predict the social image of nursing and the attitude toward nursing based on empathy, professional values and communication skills (see Figure 1). The estimated model did not present multivariate normality by Bollen-Stine bootstrap [46] (*p* = 0.010); consequently, the model was estimated using Unweighted Least Squares (ULS), which does not require this assumption. The model’s goodness-of-fit indices reflect a good fit to the data: *χ^2^*/*df* = 113.52, GFI = 0.985, NFI = 0.964, PGFI = 0.751, PNFI = 0.812, SRMR = 0.089.

The factorial weights of the five factors presented values, in most cases, higher than the minimum recommended (<0.40) by Hair et al. [32]. The three predictors showed average correlation values between 0.47 and 0.67, indicating an average association among empathy, professional values and communication skills. Regarding the predictors, linear regression weights appear in the model. According to Cohen’s f^2^ criterion [39] (where 0.14 is a low effect size, 0.36 is medium, and 0.51 is high), the predictive power is considered modest. The social image of nursing is predicted primarily by professional values (*β* = 0.41) and communication skills (*β* = 0.31); attitude toward nursing is also predicted by the same factors: professional values (*β* = 0.34) and communication skills (*β* = 0.37). Consequently, empathy emerged as a significant, though minor, predictor of the two criteria. In total, the three predictors explained a modest 30% of the variance in the social image of nursing (*R^2^* = 0.30) and 39% of the variance in attitude toward nursing (*R*^2^ = 0.39).

## 4. Discussion

This study tested a confirmatory predictive model aimed at identifying the social image of and attitude toward nursing based on empathy, professional values, and communication skills. Specifically, the three predictors collectively explain 30% of the variance in social image and 39% of the variance in attitude toward nursing. Among the predictors, professional values and communication skills significantly and positively predict a better social image of and attitude toward the profession. However, it should be acknowledged, however, that a relevant percentage of variance remains unexplained by the predictors considered in this research.

### 4.1. Image of Nursing

Our findings indicate that high levels of perceived empathy in nursing professionals contribute to consolidating a positive image of the profession, which is consistent with the findings of previous authors [47,48]. These studies posit that the social perception of nursing is directly related to the quality of care provided to individuals and families, who tend to value empathy as a defining characteristic of front-line health care professionals.

The positive correlation between working conditions (measured by the NIS) and the social image of nursing indicates that better perceived working conditions are associated with a more favorable social image. This finding is corroborated by other studies [49,50], which suggest that adverse working conditions, such as stress or professional overload, can decrease empathy levels, subsequently affecting the social perception of the profession. In environments of high care pressure, the empathy perceived by the public is reduced, thus generating a less favorable image of nursing professionals [51]. Despite advances in the professional valuation of nurses, stereotypes persist that may alter the social image of this group [52]. Consequently, promoting empathy competency in nursing curricula could improve both the quality of care and the social perception of the profession [53].

The positive and significant correlation found between professional values and both the social image of and attitude toward the profession is corroborated by different studies [54]. These studies affirm that professional nursing values, such as compassion, integrity, respect, and justice, constitute the ethical and moral basis of nursing practice, guide decision-making and interaction with individuals, and play a crucial role in social perception [55]. Participants tend to associate orientation toward ethical values with a high degree of professionalism, reinforcing the perception of nursing as an essential profession. This aligns with other research, which suggests that the social perception of nursing improves when professionals demonstrate a consistent commitment to the core values of the discipline [56]. However, Morley et al. [57] noted that incongruence between stated values and observed actions can result in dissonance in the profession’s perception. As Ozdoba highlights [58], this study shows that nursing professional values—including empathy, responsibility, and respect for individual dignity and autonomy—are the basis of daily work, and that commitment to these values is reflected in ethical and responsible care.

Communication skills are the second strongest predictor with a significant impact on the social perception of the profession. Our results indicate that clear, empathetic communication adapted to the individual needs of individuals (as measured by the EHC-PS scale) helps consolidate a positive image of nursing. Several authors have highlighted that communication skills, such as active listening, clarity in conveying information, and the ability to handle difficult conversations, are perceived as indicators of professionalism [36,59]. However, important elements in daily practice may limit the positive perception of these skills. Kwame and Petrucka [60] argue that lack of time, work overload, and cultural or language barriers can compromise the quality of communication, thereby negatively affecting the image of nursing. Communication skills training can be a powerful tool for strengthening the social perception of nursing. Specific training programs focused on improving communication skills benefit interactions with individuals and raise the status of the profession by highlighting its human and relational dimension [61].

### 4.2. Attitude Towards Nursing

As derived from the present study, professional values (such as fairness, integrity, compassion, and respect) are perceived as the essence of nursing’s commitment to individual well-being and equity in care, a finding comparable with that obtained in other studies [55,62]. When professionals exhibit strong adherence to these values, a positive attitude toward nursing is reinforced, positioning it as a trustworthy and ethical discipline [63]. Conversely, a perceived disconnect between these values and observed actions may generate negative attitudes, undermining confidence in the profession [57].

Due to the high correlation obtained between the communication sub-scales and the social image of nursing, we infer that participants tend to develop a positive view of nurses who show sensitivity and warmth in their interactions. This is consistent with previous studies, which note that effective communication reinforces trust and respect for nursing professionals [64]. However, a deficit in communication skills can lead to misunderstanding, conflict, and a negative perception of the profession, especially in high-pressure work environments [65].

Empathy is arguably the most visible component of the nurse-individual relationship; however, it demonstrated a lower direct predictive weight for social image and attitude toward the profession.

Even so, nurses’ ability to understand and share the individual’s feelings strengthens the therapeutic bond and contributes to a more satisfying care experience [66]. Individuals perceive empathetic nurses as allies in their recovery process, which significantly improves their attitude towards the profession. However, factors such as work overload or emotional exhaustion may reduce nurses’ ability to express empathy, negatively affecting the profession’s perception [67].

### 4.3. Analysis of Unstudied Variables Influencing the Social Perception of Nursing

While this study has highlighted the predictive role of empathy, professional values, and communication skills in shaping the social image of nursing, it is important to recognize additional factors that were not included in our model but may significantly affect outcomes. Among these, working conditions, organizational culture, and contextual factors have been repeatedly identified as relevant influences in the literature.

Recent high-impact studies emphasize that suboptimal working conditions—such as high workload, staff shortages, and limited resources—are key predictors of emotional exhaustion, burnout, and diminished professional performance, all of which ultimately shape public perceptions. Conversely, supportive work environments foster engagement, increase job satisfaction, and contribute to positive outcomes. Organizational culture also plays a critical role, as a positive culture not only strengthens internal cohesion but also enhances the external image of nursing through increased quality of care and public recognition. Empowering environments help nurses develop self-concept and reduce the negative effects of stereotypes.

Furthermore, contextual factors—including media representation, socio-economic environment, and exposure to crisis situations such as the COVID-19 pandemic—substantially influence the profession’s public image and social valorization. Negative stereotypes and a lack of public awareness frequently perpetuate misconceptions and undervaluation. Recent research suggests that targeted interventions in nursing management, education, and public health policy can effectively shift social norms and improve the perception of nursing in society.

The exclusion of these variables in the present study represents a limitation, signaling the need for future research to explore their mediating effects and interactions to fully understand the determinants of nursing’s social image in contemporary society.

### 4.4. Limitations

The results are based on a limited and specific data set, which may restrict their applicability to more diverse populations or different settings. Furthermore, despite its recognized importance in nursing perception, empathy had a lower predictive impact compared to professional values and communication skills. This finding may be due to the influence of variables not considered in the model, such as working conditions or emotional exhaustion. Variables such as organizational culture, work environment, and care pressure could have influenced social perception and attitudes, but were not analyzed in depth in this study.

Regarding the limitations of the snowball sampling method, as it relies on social networks and participant referrals, the method does not guarantee that the sample is random or representative of the general population. There is a high risk of self-selection bias and network bias, suggesting that the participant group may possess a higher predisposition or interest in nursing (or in responding to surveys) than the general population. 

The sample was predominantly composed of young women with higher education. This bias limits the representativeness of the results, as other demographic profiles, such as men, older people, or those with lower levels of education, may be underrepresented or absent altogether. Therefore, the interpretation and generalization of the findings should be done with caution, recognizing that the results obtained reflect to a greater extent the characteristics and perspectives of the predominant group in the sample and may not be extrapolated to the entire population of interest.

Furthermore, as this is a cross-sectional design, the obtained results reflect the specific situation of the sample at one point in time, making it impossible to state with certainty the causal influence of one variable on another. Therefore, the findings in this article may be influenced by reverse causality or uncontrolled external variables.

### 4.5. Implications for Nursing Education

Nursing education programs should prioritize the development of effective communication skills and adherence to ethical values such as fairness, integrity, and compassion. Such prioritization could improve both the social perception of and trust in the profession. It is also necessary to implement awareness campaigns that highlight the crucial role of nursing in the healthcare system, specifically by overcoming stereotypes and promoting an updated and positive image of the profession. Finally, the inclusion of modules on empathy and advanced communication skills in academic training can help prepare future professionals to face current challenges and improve the social perception of nursing.

## 5. Conclusions

Public perception of the role of nurses has shifted away from the ancillary work of administering injections and medications under medical supervision. However, establishing and maintaining a positive professional image is a long-term process that requires the joint efforts of the nursing profession and society as a whole. Based on the necessity of improving the profession’s standing, the findings of this study suggest that investing in key competencies—specifically professional values and effective communication skills—could help to drive both a better social image and a more favorable attitude toward the nursing profession.

## Figures and Tables

**Figure 1 healthcare-13-02834-f001:**
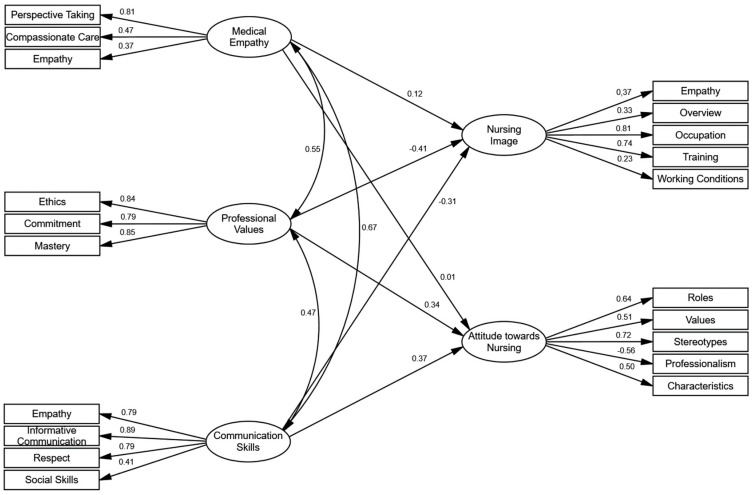
Predictive model of social image and attitude towards nursing.

**Table 1 healthcare-13-02834-t001:** Describe statistics and Pearson’s correlations for the measurements.

Measures	1	2	3	4	5	6	7	8	9	10	11	12	13	14	15	16	17	18	19	20	21	22	23	24	25
1. Working Conditions (NIS)	1.00																								
2. Training (NIS)	**0.31**	1.00																							
3. Occupation (NIS)	0.19	**0.38**	1.00																						
4. Overview (NIS)	0.04	0.14	0.24	1.00																					
5. Empathy (NIS)	0.09	0.20	**0.37**	**0.50**	1.00																				
6. Nursing Image (NIS)	0.40	**0.56**	**0.73**	**0.63**	**0.78**	1.00																			
7. Roles (NAQ)	−0.16	−0.27	**−0.38**	−0.29	**−0.46**	**−0.52**	1.00																		
8. Values (NAQ)	−0.22	**−0.32**	**−0.48**	−0.16	−0.23	**−0.44**	**0.30**	1.00																	
9. Stereotypes (NAQ)	−0.10	−0.23	**−0.43**	−0.12	−0.27	**−0.38**	**0.46**	**0.43**	1.00																
10. Professionalism (NAQ)	0.23	**0.34**	**0.31**	0.13	0.12	**0.33**	**−0.30**	−0.16	−0.21	1.00															
11. Characteristics (NAQ)	−0.04	−0.08	−0.24	−0.18	−0.27	−0.28	**0.41**	0.21	**0.46**	−0.05	1.00														
12. Nursing Attitude (NAQ)	−0.15	−0.28	**−0.49**	−0.26	**−0.44**	**−0.54**	**0.82**	**0.58**	**0.80**	−0.15	**0.64**	1.00													
13. Perspective Taking (JSE-HP)	−0.02	−0.17	−0.17	−0.03	−0.03	−0.13	0.22	0.14	0.27	−0.24	0.21	0.27	1.00												
14. Compassionate Care (JSE-HP)	−0.06	**−0.34**	−0.23	−0.06	0.03	−0.18	0.18	0.11	0.12	**−0.33**	0.10	0.14	**0.38**	1.00											
15. Empathy (JSE-HP)	−0.05	−0.22	−0.19	0.02	−0.01	−0.13	0.07	0.10	0.11	−0.16	0.05	0.09	**0.31**	**0.49**	1.00										
16. Medical Empathy (JSE-HP)	−0.06	**−0.32**	−0.26	−0.04	0.00	−0.19	0.22	0.15	0.23	**−0.34**	0.17	0.23	**0.78**	**0.84**	**0.66**	1.00									
17. Ethics (NPVS)	−0.04	−0.28	−0.27	−0.06	−0.05	−0.21	0.26	**0.30**	**0.33**	−0.27	0.12	**0.32**	**0.39**	0.25	0.12	**0.36**	1.00								
18. Commitment (NPVS)	−0.06	−0.26	−0.29	−0.08	−0.10	−0.25	0.23	**0.32**	**0.30**	−0.23	0.11	**0.31**	**0.37**	0.15	0.12	0.29	**0.65**	1.00							
19. Mastery (NPVS)	−0.08	**−0.31**	−0.26	−0.16	−0.07	−0.26	0.24	**0.31**	0.24	**−0.31**	0.11	0.28	**0.41**	0.19	0.11	**0.33**	**0.71**	**0.75**	1.00						
20. Professional Values (NPVS)	−0.07	**−0.32**	**−0.30**	−0.11	−0.08	−0.27	0.27	**0.35**	**0.32**	**−0.30**	0.13	**0.34**	**0.43**	0.22	0.13	**0.36**	**0.87**	**0.90**	**0.92**	1.00					
21. Empathy (EHC-PS)	−0.04	−0.10	−0.13	−0.04	−0.15	−0.15	0.28	0.13	**0.33**	−0.11	0.18	**0.33**	**0.44**	0.25	0.14	**0.39**	**0.33**	**0.33**	**0.30**	**0.36**	1.00				
22. Informative Communication (EHC-PS)	−0.10	−0.23	−0.24	−0.07	−0.12	−0.23	0.26	0.17	**0.31**	−0.21	0.21	**0.31**	**0.46**	**0.34**	0.27	**0.48**	**0.34**	**0.32**	**0.37**	**0.38**	**0.71**	1.00			
23. Respect (EHC-PS)	−0.07	−0.21	−0.20	−0.13	−0.18	−0.25	**0.32**	0.18	**0.40**	−0.21	0.23	**0.39**	**0.38**	0.22	0.18	**0.35**	**0.43**	**0.41**	**0.40**	**0.46**	**0.73**	**0.64**	1.00		
24. Social Skills (EHC-PS)	−0.02	−0.03	−0.10	−0.08	−0.07	−0.10	0.15	0.09	0.19	−0.06	0.10	0.18	0.24	0.04	0.15	0.18	0.14	0.17	0.20	0.19	**0.30**	**0.43**	0.25	1.00	
25. Totals Communication Skills (EHC-PS)	−0.07	−0.16	−0.20	−0.09	−0.15	−0.22	**0.31**	0.17	**0.37**	−0.17	0.22	**0.37**	**0.48**	0.27	0.23	**0.44**	**0.37**	**0.37**	**0.38**	**0.42**	**0.87**	**0.88**	**0.76**	**0.65**	1.00
M	4.11	6.74	11.31	5.69	8.51	36.35	33.55	12.98	21.32	2.51	8.12	78.47	63.06	42.00	14.27	119.33	42.91	37.27	42.15	122.33	27.09	31.63	16.58	18.01	98.67
SD	1.10	1.17	1.92	1.44	2.27	5.17	3.91	1.74	2.85	1.06	1.47	7.30	6.40	6.84	3.13	12.82	2.80	3.17	3.21	8.23	3.14	3.53	1.66	3.23	9.81
Asymmetry	0.76	1.71	0.60	0.57	0.77	0.42	−0.73	−0.75	−0.75	3.17	−0.72	−0.47	−1.11	−1.25	−0.40	−0.82	−1.46	−1.26	−1.27	−1.35	−1.31	−0.79	−1.08	−0.10	−0.69
Kurtosis	−0.20	2.65	−0.35	−0.71	−0.28	−0.32	1.81	1.24	0.41	13.19	0.27	−0.02	0.96	1.72	0.29	0.42	1.35	0.91	1.16	1.20	1.58	0.29	0.44	−0.27	0.20

Note: Correlations ≥ |±0.11| are statistically significant at 5%. Correlations ≥ |±0.30| are shown in bold.

## Data Availability

The data presented in this study are available upon request to the study’s lead author, Silvia Solera Gómez. For privacy reasons, they are not made public in this study.

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
