# Peer review of "Cross-Sectional Correlational Study in the Valencian Community (Spain) on the Social Image and Attitudes Towards Nursing"

_healthcare, 2025, doi:10.3390/healthcare13222834_

Round 1

Reviewer 1 Report

Comments and Suggestions for Authors

This study aimed to investigate a few predictors of social image and attitude toward nursing

I have the following comments:

1: The study design and location should be mentioned in the title. 

2: The study relies on a convenience snowball sample distributed via social networks, which introduces strong selection bias and limits representativeness. This should be explained in more detail in the limitations section.

3: Sample size calculation and power analysis should be presented. 

4: The cross-sectional design does not allow temporal association and causality to be examined. This should be explained in the limitations section.

5: The eligibility criteria should be described. 

6: Age, sex, and education of participants should be described in the results, not the methods. The author should explain whether this distribution represents the entire society. The demographics of the participants tend to be aggressively skewed towards women, young people, and the highly educated. This will send us back to my second comment, highlighting the lack of representativeness. The authors could have minimized this risk by stratifying the results by age, sex, and education.

7: The psychometric properties of several scales were only briefly summarized. More details should be presented.

8: The multicollinearity among predictors (empathy, professional values, and communication skills) may inflate regression coefficients, yet this issue was not assessed or discussed. The authors should statistically examine the possibility of multicollinearity.

9: The predictive power is modest (30–39% variance explained), yet the conclusions are overstated as if these predictors fully determine social image and attitudes.

10: The discussion section needs to be rewritten. The authors should focus on critical analyses of their results, make meaningful comparisons with previous studies and explain the variations per cultural perspectives, and finally give implications for policy changes. Besides, many claims in the discussion section are not supported by the results.

11: Remove highlights, main findings, and implications mentioned before the abstract. They are not part of the journal requirements. Also, remove the subheadings of the introduction and discussion sections.

12: Ethical statement mentions patients. I think the authors mean participants. Remove the ethical considerations from the methods section. It is already described in the declaration section.

13: There is overuse of authors' names in the text.

14: The limitation section is too short, given the several shortfalls of this study.

15: The introduction is too long and includes general textbook-like information. However, it did not clarify the research gap.

16: The authors provided separate theoretical frameworks guiding how empathy, values, and communication can affect social image and attitude toward nursing; yet, they did not explain how these predictors could interact.

Comments on the Quality of English Language

I recommend English language editing.

Author Response

Healthcare- Reviewer

We sincerely appreciate your insightful and thorough comments. Your feedback has been invaluable, enabling us to significantly improve the rigor, clarity, and overall quality of our manuscript. We have addressed every comment individually, implementing all suggested changes as detailed below.

Reviewer 1

This study aimed to investigate a few predictors of social image and attitude toward nursing

I have the following comments:

1: The study design and location should be mentioned in the title.

Response: The title has been revised to incorporate the study design and location, as suggested by the reviewer: "Predictive Factors of Social Image and Attitude towards Nursing: A Cross-Sectional Study in the Valencian Community, Spain."

2: The study relies on a convenience snowball sample distributed via social networks, which introduces strong selection bias and limits representativeness. This should be explained in more detail in the limitations section.

Response: We acknowledge the validity of this concern. We have added a detailed paragraph to the Limitations section of the Discussion, explaining the strong selection and network biases inherent in the snowball sampling method and discussing the resulting limitations on representativeness.

3: Sample size calculation and power analysis should be presented. 

Response: In this type of model, calculation of the sample size is often omitted because sufficiently large sizes generally offer statistically significant results for small correlation values. Therefore, we have specified the criterion of the main authors regarding sample size in the Method section: Ten participants per indicator are recommended for factor analyses [42]. In our case, we have 300 participants and 20 indicators in the most complex model, resulting in an adequate ratio of 15:1. Furthermore, we calculated the power (1 minus beta) for the correlations with a medium effect size (0.30) according to the criterion of Cohen (1992)—which have been marked in bold in Table 1—for a significance level of 5% and a sample of 300 subjects, as is our case. The power obtained was 0.998, which clearly exceeds the recommended criterion of 0.80. We have included this information in the manuscript at the end of the Method section under the Data Analysis sub-section. Reference: Cohen, J. A Power Primer. Psychol Bull 1992, 112, 155–159, doi:10.1037/0033-2909.112.1.155.

4: The cross-sectional design does not allow temporal association and causality to be examined. This should be explained in the limitations section.

Response: This information has been added to the Limitations section of the Discussion, following the reviewer’s suggestions, to clarify that the cross-sectional nature restricts the ability to infer temporal association or causality.

5: The eligibility criteria should be described. 

Response: The requested information regarding participant eligibility criteria (age, language, cognitive status) has been added to the procedure sub-section of the Method section.

6: Age, sex, and education of participants should be described in the results, not the methods. The author should explain whether this distribution represents the entire society. The demographics of the participants tend to be aggressively skewed towards women, young people, and the highly educated. This will send us back to my second comment, highlighting the lack of representativeness. The authors could have minimized this risk by stratifying the results by age, sex, and education.

Response: The authors have added an introductory paragraph in the Results section to enhance the presentation and understanding of the participant demographics, addressing the aspects highlighted by the reviewer.

7: The psychometric properties of several scales were only briefly summarized. More details should be presented.

Response: We appreciate your contribution. The scales are defined in the 'Instruments' section. All descriptions now specify their degree of internal consistency using Cronbach's alpha and detail the dimensions or factors they measure, ensuring the information is sufficiently present in the study.

8: The multicollinearity among predictors (empathy, professional values, and communication skills) may inflate regression coefficients, yet this issue was not assessed or discussed. The authors should statistically examine the possibility of multicollinearity.

Response: To assess multicollinearity between two predictors, correlations close to 0.80 must be found, following the criterion of Hair et al. (1998). As observed in Figure 1, the highest correlation among the predictors is 0.67, which is well below the critical value of 0.80. This technical discussion has been included in the Data Analysis sub-section of the Methods.

Reference: Hair, J.F.; Andreson, R.E.; Tatham, R.L.; Black, W.C. Multivariate Data Analysis; Prentice Hall, 1998; ISBN 0139305874.

9: The predictive power is modest (30–39% variance explained), yet the conclusions are overstated as if these predictors fully determine social image and attitudes.

Response: It is true that the percentage of explained variance is not high, but this is usually common in this type of non-experimental study. High values, typical of an experimental study, should not be expected. This is because not all factors that explain the two criteria considered in the model have been measured. Numerous variables not considered could explain a percentage of variance. We have addressed this in the Conclusions to qualify the results found.

10: The discussion section needs to be rewritten. The authors should focus on critical analyses of their results, make meaningful comparisons with previous studies and explain the variations per cultural perspectives, and finally give implications for policy changes. Besides, many claims in the discussion section are not supported by the results.

Response: The authors have exhaustively revised the Discussion section. It now includes the elements requested by the reviewer. While presenting a critical view of the results, findings are discussed and contrasted with those obtained by other authors in different studies, all from the dual perspective of the social image of Nursing and the attitude toward Nursing. Regarding implications, these have been addressed by including new text in the Limitations sub-section and the Conclusions of the article itself. If the reviewer believes any aspect requires further clarification, we kindly ask them to let us know.

11: Remove highlights, main findings, and implications mentioned before the abstract. They are not part of the journal requirements. Also, remove the subheadings of the introduction and discussion sections.

Response: We appreciate this comment and will proceed to remove the suggested fields (Highlights, Main Findings, etc.). As for removing the subheadings in the introduction and discussion sections, we believe that dividing both sections into subsections makes for clearer reading.

12: Ethical statement mentions patients. I think the authors mean participants. Remove the ethical considerations from the methods section. It is already described in the declaration section.

Response: We appreciate your comment. We have changed the term 'patients' to 'participants' throughout the manuscript. The Ethical Considerations have been maintained in the Methods section for clarity on the procedure, in line with journal best practice.

13: There is overuse of authors' names in the text.

Response: The authors believe that citing the names of the referenced authors alongside their corresponding citation could be appropriate and lend greater depth to the text; however, we have reviewed and modified the text as much as possible, following the reviewer’s suggestions.

14: The limitation section is too short, given the several shortfalls of this study.

Response: This section has been improved and expanded following the reviewer's instructions.

15: The introduction is too long and includes general textbook-like information. However, it did not clarify the research gap.

Response: The authors believe that these research gaps have been exposed in the Limitations sub-section, and that a significant effort has been made in the Introduction to robustly justify the study subject and back it with relevant scientific studies.

16: The authors provided separate theoretical frameworks guiding how empathy, values, and communication can affect social image and attitude toward nursing; yet, they did not explain how these predictors could interact.

Response: We find this a very interesting proposal that could undoubtedly be the subject of further analysis. The main purpose of this study was to examine the influence of these variables on the social image of nursing and their relationship to each other. Although this is a very pertinent suggestion that we will consider in future studies, the correlations between the different variables, including the predictors, are visible in Figure 1, illustrating the strength and direction of these associations.

Reviewer 2 Report

Comments and Suggestions for Authors

The manuscript addresses a relevant topic by investigating predictors of social image and attitudes toward nursing, linking empathy, communication, and professional values. The study has scientific merit and potential contribution, but it presents methodological, statistical, and editorial flaws that need to be addressed before publication is considered.

The abstract is clear, but should indicate the cross-sectional design, sampling method, and main limitations, as required by the journal's guidelines. The introduction presents a good theoretical foundation, but lacks explicitly formulated hypotheses. The methodology's use of snowball sampling, without sample power calculations, inclusion/exclusion criteria, or a defined collection period, limits reproducibility and compromises external validity. There is also an inconsistency in terminology—the text refers to "patients," even though the participants are members of the general public—which should be corrected.

The description of the instruments is insufficient: it lacks examples of items, scores, and justification for the use of short scales, such as the three-point Nursing Image Scale. The statistical analyses, performed using structural equation modeling, contain notational and interpretative errors. The reported SRMR index (0.089) is above the acceptable value, and the statement of "χ²/df = 113.52" is confusing, lacking degrees of freedom or a p-value. These aspects compromise the credibility of the results and require correction. Table 1 contains several typographical errors and numerical inconsistencies, in addition to highlighting the non-normal distribution of some variables, which would require robust analyses or polychoric correlations.

In the discussion and conclusion, the authors interpret associative relationships as if they were predictive, which is inappropriate in a cross-sectional study. It is important to reword the text to adopt more cautious language and acknowledge methodological limitations, such as selection bias, the lack of control for social desirability, and the impossibility of inferring causality. The ethics section is present, but should replace the term "patients" with "participants" and include more detailed information about the consent process. Furthermore, Healthcare recommends making data and analytical scripts publicly available, which must be complied with or duly justified.

From a formal perspective, there are numerous typographical errors, inconsistencies in tables, formatting issues, and incomplete references. Linguistic and editorial review is essential.

Comments on the Quality of English Language

The manuscript addresses a relevant topic by investigating predictors of social image and attitudes toward nursing, linking empathy, communication, and professional values. The study has scientific merit and potential contribution, but it presents methodological, statistical, and editorial flaws that need to be addressed before publication is considered.

The abstract is clear, but should indicate the cross-sectional design, sampling method, and main limitations, as required by the journal's guidelines. The introduction presents a good theoretical foundation, but lacks explicitly formulated hypotheses. The methodology's use of snowball sampling, without sample power calculations, inclusion/exclusion criteria, or a defined collection period, limits reproducibility and compromises external validity. There is also an inconsistency in terminology—the text refers to "patients," even though the participants are members of the general public—which should be corrected.

The description of the instruments is insufficient: it lacks examples of items, scores, and justification for the use of short scales, such as the three-point Nursing Image Scale. The statistical analyses, performed using structural equation modeling, contain notational and interpretative errors. The reported SRMR index (0.089) is above the acceptable value, and the statement of "χ²/df = 113.52" is confusing, lacking degrees of freedom or a p-value. These aspects compromise the credibility of the results and require correction. Table 1 contains several typographical errors and numerical inconsistencies, in addition to highlighting the non-normal distribution of some variables, which would require robust analyses or polychoric correlations.

In the discussion and conclusion, the authors interpret associative relationships as if they were predictive, which is inappropriate in a cross-sectional study. It is important to reword the text to adopt more cautious language and acknowledge methodological limitations, such as selection bias, the lack of control for social desirability, and the impossibility of inferring causality. The ethics section is present, but should replace the term "patients" with "participants" and include more detailed information about the consent process. Furthermore, Healthcare recommends making data and analytical scripts publicly available, which must be complied with or duly justified.

From a formal perspective, there are numerous typographical errors, inconsistencies in tables, formatting issues, and incomplete references. Linguistic and editorial review is essential.

Author Response

Healthcare- Reviewer

We sincerely appreciate your insightful and thorough comments. Your feedback has been invaluable, enabling us to significantly improve the rigor, clarity, and overall quality of our manuscript. We have addressed every comment individually, implementing all suggested changes as detailed below.

The manuscript addresses a relevant topic by investigating predictors of social image and attitudes toward nursing, linking empathy, communication, and professional values. The study has scientific merit and potential contribution, but it presents methodological, statistical, and editorial flaws that need to be addressed before publication is considered.

1. The abstract is clear, but should indicate the cross-sectional design, sampling method, and main limitations, as required by the journal's guidelines.

Response: The Abstract has been revised to incorporate the study design, sampling method, and main limitations, following the reviewer’s request.

2. The introduction presents a good theoretical foundation, but lacks explicitly formulated hypotheses.

Response: Following the reviewer’s recommendations, an explicit hypothesis has been added before the formulation of the objectives.

3. The methodology's use of snowball sampling, without sample power calculations, inclusion/exclusion criteria, or a defined collection period, limits reproducibility and compromises external validity.

Response: These limitations have been included at the end of the discussion section

4. There is also an inconsistency in terminology—the text refers to "patients," even though the participants are members of the general public—which should be corrected.

Response: The terminology has been reviewed and corrected throughout the manuscript.

5. The description of the instruments is insufficient: it lacks examples of items, scores, and justification for the use of short scales, such as the three-point Nursing Image Scale.

Response: The authors have included more detailed information in the Instruments section, specifying the psychometric properties, number of items, and the scoring format (Likert scale points) for each scale.

6. The statistical analyses, performed using structural equation modeling, contain notational and interpretative errors. The reported SRMR index (0.089) is above the acceptable value, and the statement of "χ²/df = 113.52" is confusing, lacking degrees of freedom or a p-value. These aspects compromise the credibility of the results and require correction. Table 1 contains several typographical errors and numerical inconsistencies, in addition to highlighting the non-normal distribution of some variables, which would require robust analyses or polychoric correlations.

Response:  We have reported the χ²/df,ratio because it is the index recommended in this type of model. The p-value, given the sample size, is always statistically significant, which is why reporting this ratio is recommended. Moreover, this ratio tends to be the worst indicator of goodness-of-fit as it often yields a very high value with large sample sizes. The other indices are more adequate. The SRMR value of 0.089 is indeed slightly higher than the criterion of 0.08, but the discrepancy is not substantial. Furthermore, the other indicators (GFI, NFI, PGFI and PNFI) show very good goodness-of-fit values. Based on this balance, we consider the model's adjustment to the data to be quite adequate. The absence of multivariate non-normality was resolved by employing the Unweighted Least Squares (ULS) estimation method, which is the recommended procedure when multivariate normality is not met, as indicated in the predictive model sub-section. Typographical errors and numerical inconsistencies in Table 1 have been corrected. 

References:

Byrne, B.M. Structural Equation Modeling With AMOS; 3rd edition.; Routledge: New York, 2016; ISBN 9781315757421.

Hair, J.F.; Andreson, R.E.; Tatham, R.L.; Black, W.C. Multivariate Data Analysis; Prentice Hall, 1998; ISBN 0139305874.

7. The ethics section is present, but should replace the term "patients" with "participants" and include more detailed information about the consent process. Furthermore, Healthcare recommends making data and analytical scripts publicly available, which must be complied with or duly justified.

 Response: The term has been changed to 'participants'. The data are available upon request to interested parties, and this justification has been added to the ethical considerations section.

8. From a formal perspective, there are numerous typographical errors, inconsistencies in tables, formatting issues, and incomplete references. Linguistic and editorial review is essential.

Response: We have performed a comprehensive linguistic and editorial review on the entire manuscript, correcting all detected typographical errors, ensuring consistency in tables and formatting, and correcting/completing all references to meet the required standards.

Round 2

Reviewer 1 Report

Comments and Suggestions for Authors

No more comments.

Author Response

Reviewer 1 has not provided any further comments on this matter. 

Reviewer 2 Report

Comments and Suggestions for Authors

The manuscript “Predictive cross-sectional study in the Valencian Community (Spain) on the social image and attitudes towards nursing” presents a relevant and current topic for nursing, investigating the influence of empathy, professional values, and communication skills on the social image and attitude of society towards the profession.
The text is well-structured, with a broad and up-to-date literature review, demonstrating theoretical mastery and good conceptual articulation. The use of validated instruments and the application of structural equation modeling (SEM) provide methodological consistency. However, there are aspects that require adjustments to ensure greater scientific rigor and interpretative clarity.
The main point of attention refers to the designation of the design as a “predictive cross-sectional study,” which is conceptually inadequate, since cross-sectional studies do not allow causal or predictive inferences in the temporal sense. It is recommended to replace it with “cross-sectional correlational study” and reformulate the conclusions to avoid deterministic language. Another limiting aspect is the snowball sampling, which resulted in a biased demographic profile (predominance of young women with high education), restricting the representativeness and generalization of the results. Although the authors acknowledge this limitation, it would be important to reinforce this discussion, making explicit the impact of this bias on the interpretations.
The results are presented clearly, with acceptable fit indices and well-described correlations, although the SRMR value slightly above the limit and the low explanatory power of the model (R² between 0.30 and 0.39) should be discussed more critically. The inclusion of a complete diagram of the structural model, with standardized coefficients, would help to visualize the relationships and strengthen analytical transparency.
The discussion is rich and consistent with the literature, but in some sections there are redundancies and causal extrapolations that need to be moderated. It would also be relevant to deepen the analysis of variables not included in the model, such as working conditions, organizational culture, and contextual factors that may affect the social perception of nursing. In formal terms, the article is well-written, in clear English, and properly organized. However, there are minor formatting errors (minor inconsistencies in references and DOI) that should be reviewed according to Healthcare's editorial guidelines.
The conclusions are aligned with the objectives, but need to employ more prudent language, emphasizing that the findings suggest—and do not demonstrate—relationships between the variables.

Author Response

We sincerely appreciate your insightful and thorough comments. Your feedback has been invaluable, enabling us to significantly improve the rigor, clarity, and overall quality of our manuscript. We have addressed every comment individually, implementing all suggested changes as detailed below.

  1. The main point of attention refers to the designation of the design as a “predictive cross-sectional study,” which is conceptually inadequate, since cross-sectional studies do not allow causal or predictive inferences in the temporal sense. It is recommended to replace it with “cross-sectional correlational study” and reformulate the conclusions to avoid deterministic language.

Response:

We have changed the concept "Predictive cross-sectional study" to "Cross-sectional correlational study in the following sections: title and abstract.

In the conclusions section, we have replaced words such as "underscore" with "suggest" and "is essential" with "could help".

  1. Another limiting aspect is the snowball sampling, which resulted in a biased demographic profile (predominance of young women with high education), restricting the representativeness and generalization of the results. Although the authors acknowledge this limitation, it would be important to reinforce this discussion, making explicit the impact of this bias on the interpretations.

Response:

In the limitations section, we have added the following paragraph to highlight the reviewer's request: “The sample was predominantly composed of young women with higher education. This bias limits the representativeness of the results, as other demographic profiles, such as men, older people, or those with lower levels of education, may be underrepresented or absent altogether. Therefore, the interpretation and generalization of the findings should be done with caution, recognizing that the results obtained reflect to a greater extent the characteristics and perspectives of the predominant group in the sample and may not be extrapolated to the entire population of interest.”

  1. The results are presented clearly, with acceptable fit indices and well-described correlations, although the SRMR value slightly above the limit and the low explanatory power of the model (R² between 0.30 and 0.39) should be discussed more critically. The inclusion of a complete diagram of the structural model, with standardized coefficients, would help to visualize the relationships and strengthen analytical transparency.

Response:

We acknowledge the reviewer's points regarding the model fit and predictive power. The goodness-of-fit indices should be considered globally; although we recognize that the SRMR value is slightly above the ideal threshold of 0.08, the remaining indices (GFI, NFI, PGFI, and PNFI) demonstrate adequate values. Therefore, we consider that the overall model fit to the data is structurally adequate. Regarding predictive power (R2), we agree that the explanatory power is modest; as indicated in the Conclusions section, we explicitly acknowledge the modesty of this value and suggest that future research is needed to verify whether this finding is replicated. Finally, Figure 1 presents the complete structural model diagram with standardized coefficients: the correlations between the latent predictors are standardized Pearson correlations and the values over the arrows leading to the two criterion variables are standardized regression β scores, a practice that ensures analytical transparency.

  1. The discussion is rich and consistent with the literature, but in some sections, there are redundancies and causal extrapolations that need to be moderated. It would also be relevant to deepen the analysis of variables not included in the model, such as working conditions, organizational culture, and contextual factors that may affect the social perception of nursing.

Response:

We have added the following in the discussion section:

Analysis of Unstudied Variables Influencing the Social Perception of Nursing

While this study has highlighted the predictive role of empathy, professional values, and communication skills in shaping the social image of nursing, it is important to recognize additional factors that were not included in our model but may significantly affect outcomes. Among these, working conditions, organizational culture, and contextual factors have been repeatedly identified as relevant influences in the literature.

Recent high-impact studies emphasize that suboptimal working conditions—such as high workload, staff shortages, and limited resources—are key predictors of emotional exhaustion, burnout, and diminished professional performance, all of which ultimately shape public perceptions. Conversely, supportive work environments foster engagement, increase job satisfaction, and contribute to positive outcomes. Organizational culture also plays a critical role, as a positive culture not only strengthens internal cohesion but enhances the external image of nursing through increased quality of care and public recognition. Empowering environments help nurses develop self-concept and reduce the negative effects of stereotypes.

Furthermore, contextual factors—including media representation, socio-economic environment, and exposure to crisis situations such as the COVID-19 pandemic—substantially influence the profession’s public image and social valorization. Negative stereotypes and lack of public awareness frequently perpetuate misconceptions and undervaluation. Recent research suggests that targeted interventions in nursing management, education, and public health policy can effectively shift social norms and improve the perception of nursing in society.

The exclusion of these variables in the present study represents a limitation, signaling the need for future research to explore their mediating effects and interactions to fully understand the determinants of nursing’s social image in contemporary society.

  1. In formal terms, the article is well-written, in clear English, and properly organized. However, there are minor formatting errors (minor inconsistencies in references and DOI) that should be reviewed according to Healthcare's editorial guidelines.

We appreciate this final confirmation of the article's overall quality and clarity. We have performed a comprehensive review of all formal and editorial elements according to Healthcare's guidelines and have corrected the issues:

  • Reference Consistency: We have meticulously reviewed all entries in the bibliography for consistency in formatting, publication details, and style.
  • Missing Details: We have corrected Reference [9] by adding the ISBN and Reference [41] by adding the ISBN and the city of publication.
  • DOI Availability: We confirm that all available DOIs have been included. For References [3] and [47], we confirm that no DOI is available, and therefore the full journal details have been provided as per journal instructions for missing DOIs.

This thorough check ensures the manuscript now adheres to all of Healthcare's editorial guidelines.

  1. The conclusions are aligned with the objectives, but need to employ more prudent language, emphasizing that the findings suggest—and do not demonstrate—relationships between the variables.

Response:  In the conclusions section, we have replaced words such as "underscore" with "suggest" and "is essential" with "could help".